# Size Effect on Hygroscopicity of Waterlogged Archaeological Wood by Simultaneous Dynamic Vapour Sorption

Liuyang Han [1] , Dehao Yu [1], Tiantian Liu [1], Xiangna Han [1,*], Guanglan Xi [1,2,*] and Hong Guo [1]

1   Institute for Cultural Heritage and History of Science & Technology, University of Science
    and Technology Beijing, Beijing 100083, China
2   National Center for Archaeology, Beijing 100013, China
*   Correspondence: jayna422@ustb.edu.cn (X.H.); xiguanglan@126.com (G.X.)

**Abstract:** Hygroscopicity is one of the most important properties of wood and plays a decisive role in its dimensional stability. In this context, conservation plans for waterlogged archaeological wood (WAW) and relevant waterlogged artefacts must be created. The size of the sample required for a moisture sorption assessment may affect the results for (and thus the perception of) the hygroscopicity of a testing artefact. Herein, to investigate the effects of the sample size on the hygroscopicity of WAW as measured via dynamic vapour sorption (DVS), typical WAW and recent (i.e., sound) wood are processed into four differently sized samples, ranging in thickness from 200 mesh to millimetre. The equilibrium moisture contents (EMCs) of the wood samples are simultaneously measured using simultaneous DVS. The sorption isotherms show that the EMC values of the recent wood at each relative humidity increase as the sample size decreases, with the superfine powder sample achieving the highest EMC of all of the recent samples. Although the WAW has a higher EMC than that of recent wood, the effect of the size of the WAW sample on its hygroscopic properties is surprisingly not as pronounced as that for the recent wood. In addition, the hysteresis between the samples of different sizes of the archaeological wood is significantly smaller than that for the reference samples. Furthermore, regarding the standard deviations of the parameters obtained from the Guggenheim Anderson de Boer and Hailwood–Horrobin models, the values for WAW are all much smaller than those for the reference wood. This further verifies the disappearance of the size effect of the hygroscopicity for WAW.

**Keywords:** waterlogged archaeological wood; size effect; simultaneous DVS; water vapour sorption; sorption model



## 1. Introduction

Wood has been widely used throughout the history of mankind. Owing to its excellent physical and mechanical properties, it has been commonly used as a building and shipbuilding material since ancient times [1–4]. Correspondingly, the investigation and protection of shipwrecks has attracted considerable interest since the successful salvage of the Vasa shipwreck in 1961 [3]. Since then, numerous shipwrecks have been excavated and conserved worldwide, such as the Mary Rose shipwreck [5], Corolla wreck [6], and Nanhai No. 1 shipwreck [7]. Each important shipwreck artefact records and carries valuable historical information regarding the construction and use of the shipwreck. The scientific evaluation of the waterlogged archaeological wood (WAW) from salvaged ancient shipwrecks can provide a reference basis for historical research on construction and navigation, as well as for the restoration and conservation of ancient ships [1,6].

As a result of prolonged submersion in the marine environment, the anatomical structure, chemical structure, and physico-mechanical properties of wood are altered. This makes WAW susceptible to deformation, cracking, and even damage during the conservation drying process [1,3]. Therefore, assessing the dimensional stability of WAW

can help in the understanding of the scientific conservation and consolidation of the wood [8–10]. The dimensional stability of a sample is typically expressed in terms of the radial, tangential, and axial linear shrinkage, cross-sectional shrinkage, and volumetric shrinkage [11,12]. As the humidity changes, the internal and external dimensions of WAW change unevenly [13]. The surface shrinkage is typically rapid, whereas the internal shrinkage is relatively slow and accompanied by stress and cracking [14,15], thus resulting in the lower dimensional stability of the WAW. The hygroscopicity of wood is closely related to its dimensional stability [16] and must be achieved to maintain the dimensional stability of the WAW.

Most previous studies on hygroscopicity primarily used two methods: a static gravimetric method using saturated salt solutions and dynamic vapour adsorption using dynamic vapour sorption (DVS) equipment [17,18]. Both these methods determine the hygroscopicity of a material based on its equilibrium moisture content (EMC). However, the saturated salt solution method is time-consuming and labour-intensive, with limited accuracy [18,19]. The conventional DVS instrument was invented in the 1980s and obtains high-accuracy data, involving less time and effort [19,20]. However, it can test only one or two samples per test, potentially leading to the risk of testing errors during the multiple measurements for different samples [13]. Recently, researchers have investigated the relationship between WAW and hygroscopicity using a technique called simultaneous DVS, which can accurately measure up to 23 wood samples simultaneously [13,21]. The results from experiments are generally more convincing when more samples are simultaneously measured under the same environmental conditions [21]. In addition, various sample sizes can be used in different DVS tests in different studies, such as superfine powders [22], powders [23,24], ground 0.5–1.0-mm particles [25], millimetre-thick strips [16], $4 \times 4 \times 1$ mm$^3$ flakes [26], $15 \times 4 \times 0.5$ mm$^3$ sticks [8,27], and centimetre-thick wood blocks [28]. However, researchers have not yet studied whether the results are comparable with different WAW sample sizes. If these results are not comparable, the sampling and testing criteria for WAW must be standardized to avoid wasting valuable samples. If these results are comparable, researchers can focus less on the effects of the sample size on the results, and instead increase the efficiency of the relevant research and conservation programmes. Therefore, research on the effect of size on the hygroscopicity of WAW is important and urgent.

This study examined a waterlogged plank from the Shengbeiyu shipwreck and sound reference wood. Samples of different sizes were simultaneously tested through simultaneous DVS. In addition to analysing the equilibrium moisture content and hysteresis of samples of different sizes, the sorption isotherms obtained were fitted to the Hailwood–Horrobin (H–H) model and the Guggenheim Anderson de Boer (GAB) model for further analysis. The size effect on the hygroscopicity of WAW was investigated to provide guidance for future sampling, evaluation and conservation of WAW, and related waterlogged artefacts.

## 2. Materials and Methods

### 2.1. Materials

The WAW sample in this study was selected from one of two planks retrieved from the Shengbeiyu shipwreck during an underwater survey in 2021 [13]. First, the same part from the visually sound region of the plank was evenly cut with a blade into four equal groups in the shape of blocks (Figure 1), with each group having a wet weight of approximately 200 mg. All of the sample groups were air-dried for at least one month in the experimental environment and were then vacuum-freeze-dried before further processing. Next, the first group was milled into a 200 mesh superfine powder using the EFM Freezer Mill 6770 (SPEX SamplePrep, Metuchen, NJ, USA) (this group was named Wa). The second group was milled into a 60–80 mesh powder using a coffee grinder (this group was denoted Wb). The third group was cut into approximately 15-μm-thick slices using a microtome (Leica Autocut) (Wc), and the fourth group was cut into millimetre-thick strips (Wd). The reference wood (FR) was selected according to the identification result for the WAW and the

FR samples were prepared using the same processes as in the comparable WAW samples; these samples were named Fa, Fb, Fc and Fd, respectively (Figure 1).

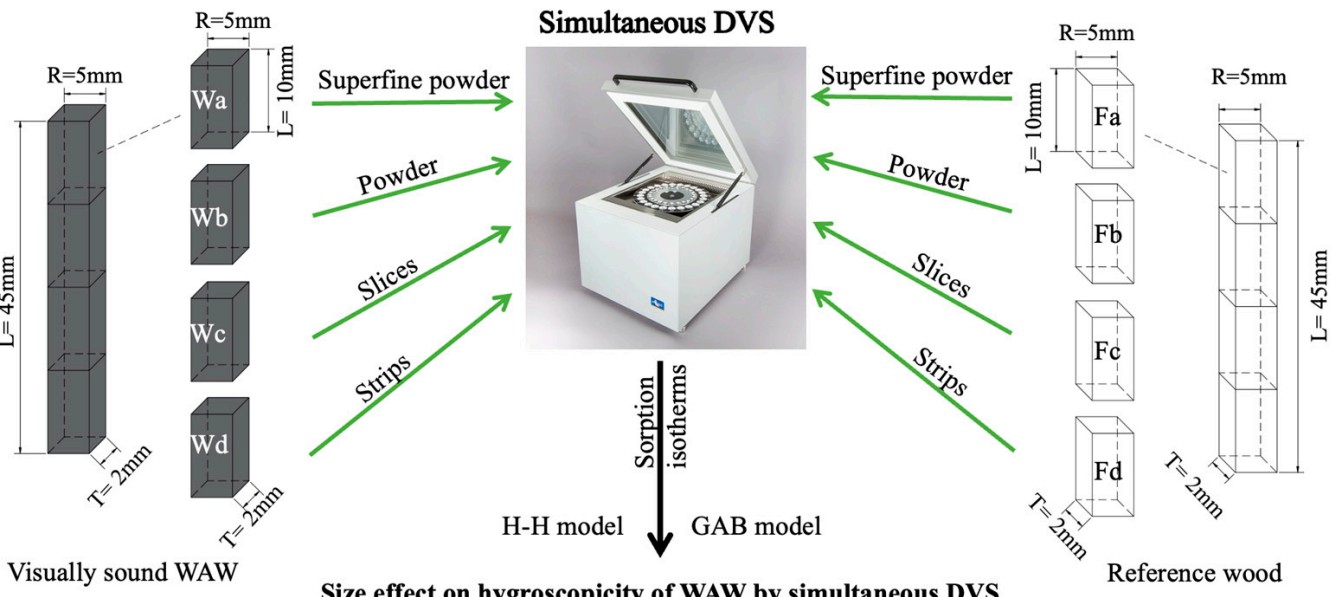

**Figure 1.** Scheme for the preparation and the testing of the DVS samples.

### 2.2. Methods

#### 2.2.1. Wood Identification

The samples (cubes with 0.5-cm side lengths) for wood identification were cut from the plank of the Shengbeiyu shipwreck. The general process of wood identification was based on previously published papers [1,13] and can be summarised as follows. First, the samples (thickness, a. 15 µm) were cut by a slicer (Leica Autocut, Germany). Next, slices for wood identification were prepared by following a general process: dyeing, dehydration, transparency, and sealing [1]. Subsequently, the anatomical structures were observed using an optical microscope (BX 50, Japan). Finally, the anatomical structures were compared by referring to the IAWA's list for softwoods [1,7].

#### 2.2.2. Maximum Water Content (MWC) and Basic Density (BD)

MWC and BD of WAW samples can be used for assessing the degradation degree for WAW [29–31]. The mass of three parallel WAW samples (cube with 0.5 cm side length) was measured with an electronic balance and the value recorded as $m_1$ before drying at $103 \pm 2\,°C$ in an oven and being reweighed and recorded as $m_0$. The drainage method was applied to measure the volume $V_1$ of the WAW samples as previously reported [7]. The MWC and BD of the samples were obtained according to Formulas (1) and (2):

$$\text{MWC} = \frac{m_1 - m_0}{m_0} \times 100\% \tag{1}$$

$$\text{BD} = \frac{m_0}{V_1} \times 100\% \tag{2}$$

#### 2.2.3. Simultaneous DVS

The EMCs of the WAW and reference wood at various steps of relative humidity (RH) were measured via simultaneous DVS (SPSx, Germany). The measurements can be summarised as follows. The samples were exposed to ascending RH steps ranging from 0 to 95% (intervals of 10% RH from 0% to 90%) during the adsorption process at 25 °C and

then descending in the same manner during desorption process. The equilibrium in each step was set as less than 0.0001%/min for a mass change per time (dm/dt).

### 2.2.4. Sorption Models

The following two classical models were fitted to further illustrate the hygroscopicity of WAW and reference wood (Figure 1).

### The GAB Model

The parameters of the GAB model were calculated as follows with Origin 2022 software (OriginLab Corporation, USA) [16].

$$EMC = \frac{RH \cdot M_m \cdot C_{GAB} \cdot K_{GAB}}{(1 - K_{GAB} \bullet RH) \bullet (1 - K_{GAB} \cdot RH + C_{GAB} \cdot K_{GAB} \cdot RH)} \times 100\% \qquad (3)$$

where $RH$ (%) is the relative humidity of the air; $M_m$ is the monolayer capacity; $C_{GAB}$ (%) is an equilibrium constant related to monolayer sorption; and $K_{GAB}$ (%) is an equilibrium constant related to multilayer sorption [16].

### The H–H Model

The H–H model equation is as follows [13].

$$EMC = M_h + M_s = \frac{1800}{w} \cdot \frac{k_1 \cdot k_2 \cdot RH}{100 + k_1 \cdot k_2 \cdot RH} + \frac{1800}{w} \cdot \frac{k_2 \cdot RH}{100 - k_2 \cdot RH} \times 100\% \qquad (4)$$

where $EMC$ (g/g) is the $EMC$; $RH$ (%) is the $RH$; $M_h$ is the monolayer moisture content (%); $M_s$ is the multilayer moisture content (%); $w$ is the molecular weight of the wood at every adsorption site; and $k_1$ and $k_2$ are equilibrium constants in the sorption process [13,32].

## 3. Results and Discussions

### 3.1. Wood Identification of Waterlogged Archaeological Wood (WAW)

To select a recent wood for the reference sample with the same species as the WAW, a wood identification study was conducted on the WAW. Figure 1 depicts the light microscopy results from the sample from the shipwreck plank used in this study. The wood plank from the shipwreck was microscopically identified as *Cunninghamia* sp. (Chinese fir), occasionally consistent with our previous study on other planks recovered from the Shengbeiyu shipwreck during an underwater survey in 2021 [13]. Based on the wood identification results, a reference Chinese fir wood from Nanping City in China was selected. The anatomical characteristics of *Cunninghamia* sp. can be briefly described as follows: a growth ring with an early wood to late wood gradient, with a cross-section of early wood tracheids of irregular polygons and squares and that of late wood tracheids of rectangles and polygons; tracheid-bordered pitting in the radial walls in a single or (occasionally) double row, with cross-field pitting cryptomeripsoides; a ray width of one cell and ray height of 2–8 cells [13].

MWC and BD of the WAW sample were $537 \pm 17\%$ and $0.17 \pm 0.02$ g/cm$^3$, respectively; thus, according to previous studies, the sample was classified as severely decayed wood (MWC, >400%) [2,33]. Notably, the coefficients of variation corresponding to the MWC and BD were both smaller than those in the previous report [13], thus indicating that the degradation of the plank selected for this study was more uniform. This also indicated that the variability between the samples in the different groups in this study was low, thereby ensuring that the differences in sorption behaviours between the different groups in our subsequent study were owing to the sample size alone. For the reference wood, MWC and BD were $151 \pm 14\%$ and $0.46 \pm 0.02$ g/cm$^3$, respectively.

*3.2. Sorption Isotherms*

The relationship between the wood EMC and ambient RH at a constant temperature (T) is described as a sorption isotherm [20]; this represents the most basic and important data for studying the behaviours of moisture sorption. The sorption isotherms from 0% to 98% RH at 25 °C for the reference Chinese fir and WAW samples in different sizes can be seen in Figure 2. Evidently, S-shapes were observed in all of the adsorption and desorption plots, thus suggesting that sorption isotherms of the archaeological and reference Chinese fir samples of different sizes reflect IUPAC Type II pattern isotherms [27,34,35]. A comparison of Figure 2A,B reveals that the EMC of the severely degraded WAW was always higher than that of sound wood for the same size at each RH, thus being consistent with our previous finding that a WAW exhibits a much higher EMC value than that of the recent wood [13].

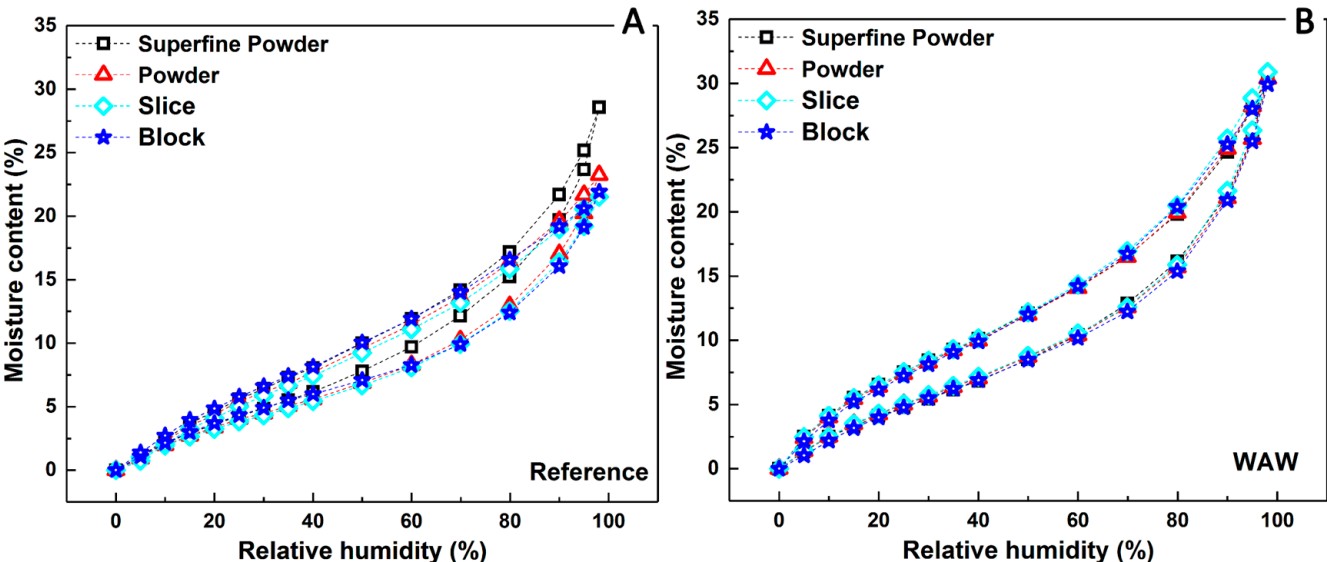

**Figure 2.** Equilibrium moisture content (EMC) sorption isotherms of the reference samples (**A**) and WAW samples (**B**) versus relative humidity (RH) ranging from 0 to 98%.

Figure 2A shows that the hygroscopicity for the four samples of the sound recent wood increased in the order of: superfine powder sample (Fa) > powder sample (Fb) > block sample (Fd) ≥ slice sample (Fc). The EMCs of the four samples were similar in the humidity range below 30% RH, and their maximum EMCs during adsorption were 4.85, 4.47, 4.31, and 4.90%, respectively. The EMCs of the samples increased dramatically above 60% RH, which is in agreement with the results of previous studies [13,36]. Further, the EMCs of the superfine powder sample became increasingly higher than those of the other samples as the RH increased above 70% during the hygroscopic process. Consequently, the EMCs of Fa, Fb, Fc, and Fd reached their maximum values of 28.58, 23.25, 21.52, and 21.91% at the highest RH of 98%, respectively. These results suggest that the hygroscopicity of the samples increased as the particle size decreased (from 200 mesh to millimetre thickness) in the DVS test. This may be owing to the increased specific surface area and adsorption sites of the samples, which are beneficial for the sorption of the multilayer moisture content [22]. The above results also prove that the size of the wood sample influences its hygroscopicity.

Figure 2B shows that the four WAW samples exhibited almost the same hygroscopic behaviours in the sorption isotherms. In particular, the EMCs of the four WAW samples were similar in the humidity range below 30% RH. At this range, their maximum EMCs during adsorption were 5.44, 5.64, 5.75, and 5.50%, respectively. The EMCs of the samples increased dramatically above 60% RH, similar to the case of the recent wood. However, the EMCs of the superfine powder sample did not increase significantly above 70% RH compared with the other three samples during the hygroscopic process. Even at the highest RH, the EMCs of Wa, Wb, Wc, and Wd were almost equal (30.64, 30.38, 30.89, and 29.93%,

respectively). Furthermore, the standard deviation of the EMCs between the WAW samples at each RH was generally less than 50% of that of the reference samples (Figure 2B, Table S1). This suggests that the relationship between the hygroscopicity of the WAW samples and their sizes (from 200 mesh to millimetre thick) is not as pronounced as that for a sound recent wood.

### 3.3. Hysteresis

In addition to the sorption isotherms, the sorption hysteresis values also differ between the recent and WAW woods. Herein, hysteresis is defined as the difference in the EMC values between desorption and adsorption at the same RH. It can be used to describe the incomplete reversibility of the sorption when water molecules enter and leave the cell wall matrix in plant fibres [22]. Evidently, all WAW samples exhibited higher hysteresis than those of reference samples throughout the entire investigated RH range (Figure 3), which is probably owing to the higher proportions of amorphous areas in the WAW [13]. Furthermore, the hysteresis of all samples tended to increase with increasing RH, particularly in the RH range above 50%. In addition, the hysteresis between the WAW samples with different sizes was significantly smaller than that between the reference samples (Figure S1), thus suggesting that the relationship between the hygroscopic behaviour of a WAW sample and its size (from 200 mesh to millimetre thick) is not as evident as that in recent wood.

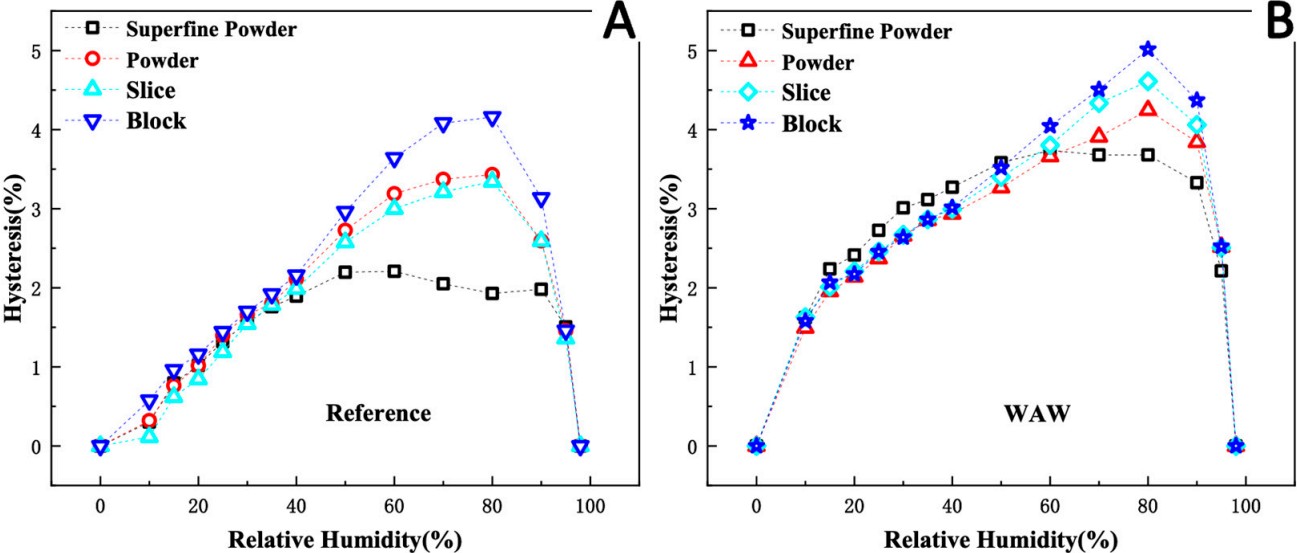

**Figure 3.** Hysteresis values of reference samples (**A**) and WAW samples (**B**) versus RH.

### 3.4. Sorption Model

To further illustrate the influence of the sample size on the hygroscopicity of WAW and sound recent wood, classical GAB and H–H models were applied to fit the adsorption and desorption isotherms of the WAW samples Wa, Wb, Wc, and Wd and recent wood samples Fa, Fb, Fc, and Fd. The fits were considered valid when all of the values of the coefficient of determination ($R^2$) exceeded 0.99 [16,33,37]. The parameters calculated using the least squares fit are listed in Table 1.

**Table 1.** Coefficients of the Guggenheim Anderson de Boer (GAB) and Hailwood–Horrobin (H–H) models for waterlogged archaeological wood and recent wood.

| Process | Sample | GAB Model | | | | | | H–H Model | | | |
|---|---|---|---|---|---|---|---|---|---|---|---|
| | | $R^2$ | $M_m$ | $C_{GAB}$ | $K_{GAB}$ | $S_{GAB}$ | $w$ | $k_1$ | $k_2$ | $R^2$ | $p$ |
| adsorption | Fa | 0.998 | 5.57 | 6.12 | 0.82 | 212.44 | 316.38 | 4.84 | 0.82 | 0.999 | 3.16 |
| | Fb | 1 | 4.91 | 6.98 | 0.81 | 187.27 | 367.24 | 6.04 | 0.81 | 1 | 2.72 |
| | Fc | 1 | 5.01 | 6.28 | 0.79 | 191.16 | 361.34 | 5.38 | 0.79 | 1 | 2.77 |
| | Fd | 0.999 | 4.80 | 9.44 | 0.80 | 183.30 | 374.08 | 8.52 | 0.80 | 0.999 | 2.67 |
| | Wa | 0.999 | 5.86 | 7.20 | 0.82 | 223.69 | 304.34 | 6.07 | 0.82 | 0.999 | 3.29 |
| | Wb | 0.999 | 5.69 | 8.57 | 0.83 | 217.13 | 313.69 | 7.47 | 0.83 | 0.999 | 3.19 |
| | Wc | 0.999 | 5.77 | 8.44 | 0.83 | 220.15 | 311.61 | 7.47 | 0.83 | 0.999 | 3.21 |
| | Wd | 0.999 | 5.72 | 7.23 | 0.83 | 218.16 | 314.91 | 6.26 | 0.83 | 0.999 | 3.18 |
| desorption | Fa | 0.999 | 7.59 | 6.14 | 0.76 | / | 233.21 | 4.95 | 0.75 | 0.999 | / |
| | Fb | 1 | 9.45 | 4.45 | 0.65 | / | 190.30 | 3.44 | 0.65 | 0.999 | / |
| | Fc | 0.999 | 10.72 | 3.63 | 0.60 | / | 168.84 | 2.65 | 0.60 | 0.999 | / |
| | Fd | 1 | 10.56 | 4.97 | 0.59 | / | 171.07 | 3.98 | 0.59 | 0.999 | / |
| | Wa | 1 | 8.84 | 10.06 | 0.73 | / | 203.76 | 9.09 | 0.73 | 0.999 | / |
| | Wb | 1 | 8.98 | 9.21 | 0.73 | / | 201.42 | 8.31 | 0.73 | 1 | / |
| | Wc | 1 | 9.30 | 8.89 | 0.73 | / | 195.70 | 7.99 | 0.73 | 0.999 | / |
| | Wd | 1 | 9.80 | 7.14 | 0.71 | / | 186.61 | 6.35 | 0.71 | 0.999 | / |

Note: $M_m$ is the monolayer capacity, $C_{GAB}$ is the equilibrium constant associated with monolayer sorption, $K_{GAB}$ is the equilibrium constant related to multilayer sorption, $S_{GAB}$ (m$^2$/g) is the internal specific surface area [16]. $w$ is the molecular weight of wood at every adsorption site, $k_1$ and $k_2$ are equilibrium constants in the sorption process, $M_h$ is the monolayer moisture content (%), $M_s$ is the multilayer moisture content (%) [13,33]. $p$ represents the number of adsorption sites in wood, mainly hydrophilic groups (–OH and C=O) [38].

### 3.4.1. GAB Model

The fitting parameters in the theoretical GAB model exhibited a high fitting accuracy ($R^2 > 0.99$) [16,33,37], thus indicating that the GAB model is highly suitable for understanding the relationship between the RH and EMC for the recent wood and WAW. At the adsorption stage, the $M_m$ values of the four WAW samples were 5.86, 5.69, 5.77, and 5.72, respectively, which are all higher than those of the reference sample of the same size with values of 5.57, 4.91, 5.01, and 4.80, respectively. This proves that the size effect on the amount of monolayer sorption sites of the WAW sample is much smaller than that on the reference samples. Notably, the $M_m$ values of the reference sample ranged from 4.81 to 5.57%, whereas those of the WAW sample ranged from 5.69 to 5.86%; in addition, the standard deviation of the $M_m$ value of the WAW samples was only 22.23% of that of the reference samples in the adsorption stage (Table S2). The highest $M_m$ was obtained from the superfine powder samples among all sizes of both the reference and WAW samples (5.57% and 5.86%, respectively). Considering that a higher $M_m$ is associated with a lower crystallinity index [39] (related to the degree of order and crystal size [40] of the cell wall substance [41]), milling was considered to somehow reduce the crystallinity of superfine powder samples [39]. The parameter $K_{GAB}$ is a correction factor for multilayer molecules and is calculated relative to the volume of the liquid. When $K_{GAB}$ is close to 1, the molecules outside the monolayer have the same properties as pure water [22,42]. For the reference samples, the $K_{GAB}$ values of samples Fa and Fb were higher than those of the other two samples, thus indicating that the water in these two samples is closer to free water than in Fc and Fd. In contrast, the $K_{GAB}$ values of the WAW samples exhibited a striking similarity, with only 12.29% of the standard deviation of those of the reference samples (Table S2). Parameter $S$ can be used to compare the monolayer sorption capacities of the samples. The variability in $S$ values between WAW samples of different sizes was smaller than that of the reference samples (Table 1). The standard deviation of $S$ value of the WAW samples was approximately one-fifth of the standard deviation of the reference samples (Table S2).

For the desorption stage, the $M_m$ values for the reference samples and WAW samples ranged from 7.59 to 10.72% and from 8.85 to 9.80%, respectively. Meanwhile, the superfine

powder samples had the smallest $M_m$ values among all samples. In addition, the standard deviation of the S values of the WAW samples was only one-third of that of the reference samples. The above GAB model fitting results indicate that the size (from 200 mesh to millimetre thickness) has an effect on the sorption behaviours of the sound wood samples; however, this effect is not significant for the WAW samples.

### 3.4.2. H–H Model

The fitting parameters in the H–H model (i.e., $w$, $k_1$, and $k_2$) exhibited a high fitting accuracy ($R^2 > 0.99$) for all samples, thus indicating that the H–H model can effectively describe the experimental sorption data [13,16,33] (Table 1). $p$ represents the number of adsorption sites in the wood, primarily hydrophilic groups (–OH and C=O). The highest $p$ value among the reference samples was obtained for the superfine powder sample. The dispersion of $p$ values of the WAW samples was smaller than that of the reference samples. The standard deviation of the S values of the WAW samples was only 22.18% of that of the reference samples (Table S2). Thus, the influence of the size (from 200 mesh to millimetre thickness) on the number of monolayer sorption sites of the WAW sample was smaller than that on the sound wood sample (this was also verified through the previous inference).

According to the H–H model, the total adsorbed water content can be divided into the monolayer moisture content ($M_h$) and multilayer moisture content ($M_s$) [41,43]. Figure 4 shows the $M_h$ and $M_s$ values during the adsorption processes of the reference samples (Figure 4A,C) and WAW samples (Figure 4B,D). Evidently, the monolayer moisture content initially increased dramatically in the relatively low humidity range (0–30% RH) and then stabilised at 40% RH (Figure 4A,B). However, the values of the recent wood samples of different sizes at each RH were evidently different (Figure 4A), whereas those of the WAW samples were similar (Figure 4B). Immediately afterward, in the humidity range of 40–98%, the highest $M_h$ values of the four reference samples were 4.52, 4.05, 4.02, and 4.18, respectively, whereas those of the corresponding WAW samples were similar (4.91, 4.92, 4.96, and 4.77, respectively). The standard deviation of the WAW samples at 98% RH was only 13.75% of that of the reference samples. This indicates a reduced effect from the size on the hygroscopicity of the WAW in terms of the monolayer moisture sorption.

The $M_s$ values, which reflect the multilayer moisture content, increased with increasing of RH; the rate of increase also increased with the RH from 0 to 98% RH. When the RH was lower than 40%, the $M_s$ values of Fa, Fb, Fc, and Fd were below 2.76, 2.34, 2.31, and 2.25%, respectively, whereas those of Wa, Wb, Wc, and Wd were below 2.90, 2.84, 2.87, and 2.82%, respectively. Although the WAW samples exhibited Ms curves similar to those of the reference samples at RH values below 40%, the $M_s$ values for RH values above 40% were evidently higher than those of the reference samples (Figure 4C,D). Taking 98% RH as an example, the Ms values of the four reference samples were 23.01, 18.86, 17.39, and 17.26%, respectively, whereas those of the relevant WAW samples were 24.87, 24.56, 25.21, and 24.37%, respectively. In addition, the $M_h$ and $M_s$ curves of the WAW samples of different sizes were very close to each other, whereas the curves of the reference samples of different sizes were more significantly different, particularly at high RH ($\geq$70% RH). This can be attributed to the standard deviations of the samples of different sizes (Tables S3 and S4).

Figure 5 shows the $M_h$ and $M_s$ values of the reference and WAW samples during the desorption process. Evidently, similar to the adsorption process, the $M_h$ and $M_s$ values of the reference samples were lower than those of the WAW samples. Furthermore, greater differences were observed between the $M_h$ and $M_s$ curves for the reference samples of different sizes than for the WAW samples (Tables S3 and S4). Above all, the H–H model confirms the disappearance of the size effect of the hygroscopicity for WAW at the levels of the monolayer moisture sorption and multilayer moisture sorption behaviours.

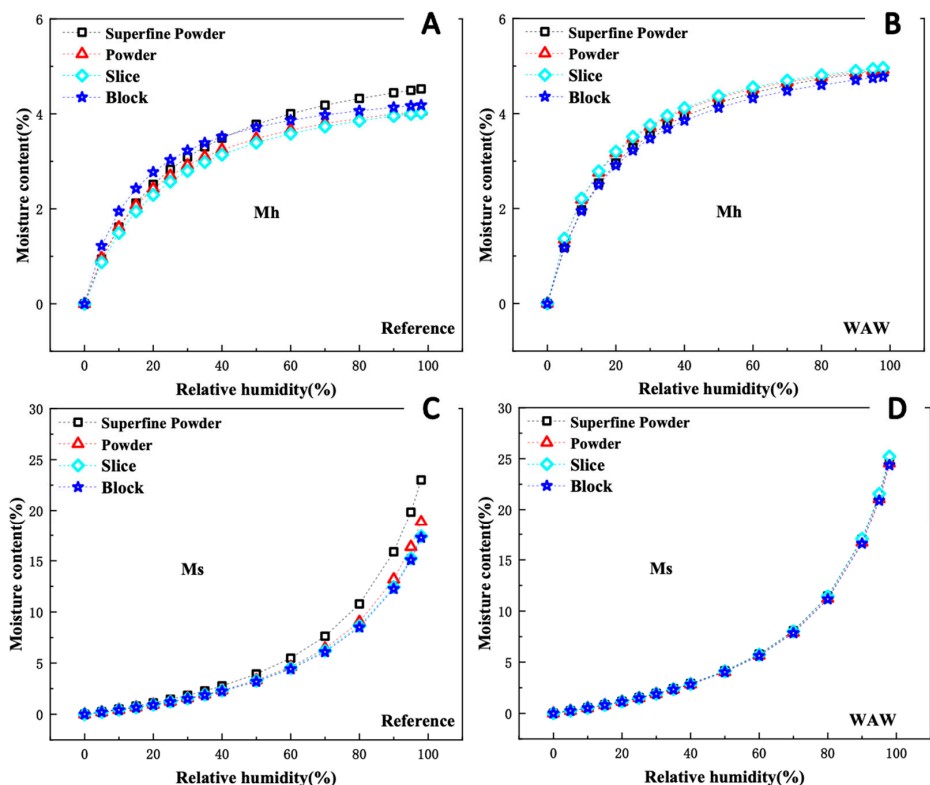

**Figure 4.** $M_h$ of reference samples (**A**) and WAW samples (**B**) in adsorption process. $M_s$ of reference samples (**C**) and WAW samples (**D**) in adsorption process.

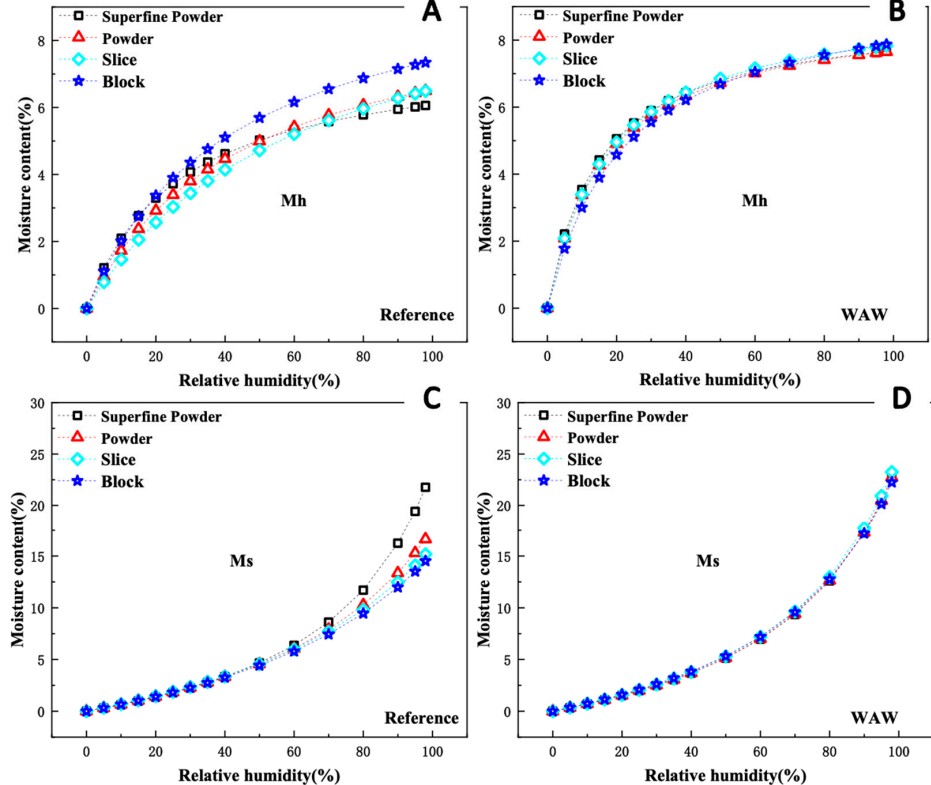

**Figure 5.** $M_h$ of reference samples (**A**) and WAW samples (**B**) in desorption process. $M_s$ of reference samples (**C**) and WAW samples (**D**) in desorption process.

## 4. Conclusions

The sorption isotherms obtained via simultaneous DVS in this study revealed that the EMC values of WAW samples were higher than those of sound wood samples with sizes ranging from 200 mesh to millimetre thickness at each RH. More importantly, the relationship between the hygroscopic behaviour of the WAW sample and its size was not as evident as that for recent wood. The fitting results from the classical GAB and H–H models revealed that the WAW samples of different sizes had similar numbers of adsorption sites and the corresponding standard deviations of the parameters of the WAW samples were also similar to each other, thus indicating that WAW samples of different sizes have similar hygroscopicity and sorption behaviours. The DVS test results were comparable between severely degraded WAW samples of different sizes. This may increase the efficiency of relevant research and conservation programmes by providing guidance for the future sampling, assessment, and conservation of waterlogged artefacts. For sound wood, the size of the sample affects the hygroscopicity, particularly for small samples.

Future work should introduce multidisciplinary methods to further investigate the reason(s) for the size effect on the hygroscopicity of recent wood and disappearance of this effect in WAW. In addition, attempts should be made to increase the sizes of the examined samples (e.g., by improving equipment or testing techniques) from millimetre to centimetre thickness to provide more practical basic data for the conservation of waterlogged wooden artefacts.

**Supplementary Materials:** The following supporting information can be downloaded at: https://www.mdpi.com/article/10.3390/f14030519/s1, Figure S1: Hysteresis for Reference samples and WAW samples versus RH ranged from 50% to 90%; Table S1: Standard deviation of EMC of reference samples and WAW samples size at different RH in adsorption process and desorption process; Table S2: Standard deviation of parameters for models; Table S3: Standard deviation of Mh value of reference samples and WAW samples size at different RH in adsorption process and desorption process; Table S4: Standard deviation of Ms value of reference samples and WAW samples size at different RH in adsorption process and desorption process.

**Author Contributions:** Conceptualization, L.H. and G.X.; methodology, L.H. and H.G.; software, L.H. and D.Y.; validation, L.H., G.X. and H.G.; formal analysis, L.H.; investigation, L.H., T.L.; resources, G.X.; data curation, L.H.; writing—original draft preparation, L.H. and D.Y.; writing—review and editing, L.H., H.G. and G.X.; visualization, L.H.; supervision, X.H., H.G. and G.X.; project administration, L.H., H.G. and G.X.; funding acquisition, L.H. and G.X. All authors have read and agreed to the published version of the manuscript.

**Funding:** This research was funded by the National Key Research and Development Program of China (Grant Nos. 2020YFC1521801, 2020YFC1521804), and fundamental research funds for the central universities (University of Science and Technology Beijing) (Grant No. FRF-TP-20-102A1).

**Data Availability Statement:** The data presented in this study are available on request from the corresponding author.

**Acknowledgments:** The authors would like to thank Yafang Yin from Chinese Academy of Forestry for comments and suggestions. The authors also thank Xiaomei Jiang and Yonggang Zhang from the Chinese Academy of Forestry for the wood identification support.

**Conflicts of Interest:** The authors declare no conflict of interest.

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
