# Peer review of "Size Effect on Hygroscopicity of Waterlogged Archaeological Wood by Simultaneous Dynamic Vapour Sorption"

_forests, doi:10.3390/f14030519_

Round 1
Reviewer 1 Report
The paper brings new information about hygroscopicity of waterlogged archaeological wood and represent a part of an extensive study about this subject of the authors.
The results are useful in conservation field of the artefacts. The manuscript is well written, easy to read and understand. Results are supported by statistical interpretation and data.
However, I suggest some clarifications:
In section Method
I suggest to explain the protocol for selection and preparation of the samples to be used for this research before milling. How did you clean the wood before use? Did you take in consideration the degradation state of the wood or you considered an apparently sound zone or randomly??
Indicate the number of the samples tested for each group. It is 23 as previously presented in other work [ref. 13]?
You repeat/copy the methodology for sections 2.2.1, 2.2.2, 2.2.3.2 as your previous work [ref. 13]. Please, shorten it. Maybe microscopy should no longer be presented in this paper since it appears in the other publication [13]. It’s the same result.
Results
3.1- Wood identification
Please revise the wood species. Cunninghamia sp. is not a fir wood. It belongs to Cupressaceae family not Pinaceae as fir. Maybe you explain if it is known as Chinese fir or is a regional name??
In situation you don’t remove the microscopy, in Figure 1 you have to indicate the anatomic characteristics on the figures.
Generally, I suggest to eliminate the section Microscopy and wood identification which repeats as previous work and maybe, you could add some results about density and some correlations with moisture content.
Author Response
Dear reviewer,
We appreciate your interest in our work (manuscript number: forests-2228141) and are happy to consider any comments.
We have tried our best to answer your questions/comments, and we have done so according to the change in the revised manuscript marked up using the "Track Changes" function.
This manuscript has been improved by your professional comments and suggestions.
Please refer to the attached file for details.
Bests,
The authors.

Reviewer 2 Report
Dear authors:
Your study is well designed, the analysis is thorough and the paper is well written and easy to read.
In referring to Table S1 in line 216, you wrote Table. S1 rather than Table S1. Check for similar typo errors in line 261, 272, 287, and 292.
There are two occurrences of Section 2.2.2.
In Section 2.2.2 on the determination of maximum water content and basic density, you indicates how humid volume m1 was measure. However, it is not clear how the samples were saturated to get the fully saturated samples.
In Section 2.2.2 (lines 127-128), you wrote “the samples (approximately 25 mg) were cut into millimetre-thick stripes using a sharp knife”. I believe that the description stands only for Group 4 samples. You may need to add the protocol for other groups as well.
In line 168, you reported values of maximum water content and basic density for waterlogged wood. For comparison purpose, I would recommend indicating the maximum water content and the basic density for sound wood too.
In line 201, you wrote: “These results suggest that that the hygroscopicity of the samples increased as the particle size decreased.” While this may be partially true given the increase specific surface area and the availability of sorption sites in small samples, I do believe that a large part of the discrepancy between small and large samples is due to the lack of equilibrium for large samples. By extending the duration of the tests for large samples, the results could be different. As such it is hard to conclude that the size has a significant effect on the hygroscopicity of sound wood.
Author Response
Dear reviewer:
We very much appreciate your interest in our work (manuscript number: forests-2228141) and are happy to consider each of your comments. Below we have tried our best to answer your questions/comments according to the change in the revised manuscript, which has been marked using the "Track Changes" function.
We believe that this manuscript has been improved by your professional comments and guidance.
Thank you very much for your kind help!
Yours sincerely,
The Authors

Round 2
Reviewer 1 Report
The manuscript was improved as suggested recommendations.
The protocol for sample preparation is now explained and results are argued with statistical data.
The article brings a relevant contribution for wooden artefacts conservation .